# Metastatic Melanoma: Liquid Biopsy as a New Precision Medicine Approach

**DOI:** 10.3390/ijms24044014

**Published:** 2023-02-16

**Authors:** Elena Ricciardi, Elena Giordani, Giovanna Ziccheddu, Italia Falcone, Patrizio Giacomini, Maurizio Fanciulli, Michelangelo Russillo, Marianna Cerro, Gennaro Ciliberto, Aldo Morrone, Antonino Guerrisi, Fabio Valenti

**Affiliations:** 1UOC Oncological Translational Research, IRCCS-Regina Elena National Cancer Institute, 00144 Rome, Italy; 2SAFU, Department of Research, Advanced Diagnostics, and Technological Innovation, IRCCS-Regina Elena National Cancer Institute, 00144 Rome, Italy; 3Clinical Trial Center, IRCSS-Regina Elena National Cancer Institute/San Gallicano Dermatological Institute, 00144 Rome, Italy; 4Division of Medical Oncology 1, IRCCS-Regina Elena National Cancer Institute, 00144 Rome, Italy; 5Department of Clinical and Molecular Medicine, “La Sapienza”, University of Rome, 00185 Rome, Italy; 6Scientific Direction, IRCCS-Regina Elena National Cancer Institute, 00144 Rome, Italy; 7Scientific Direction, San Gallicano Dermatological Institute IRCCS, 00144 Rome, Italy; 8Radiology and Diagnostic Imaging Unit, Department of Clinical and Dermatological Research, San Gallicano Dermatological Institute IRCCS, 00144 Rome, Italy

**Keywords:** liquid biopsy, metastatic melanoma, precision medicine, circulating tumor cells, circulating tumor DNA

## Abstract

Precision medicine has driven a major change in the treatment of many forms of cancer. The discovery that each patient is different and each tumor mass has its own characteristics has shifted the focus of basic and clinical research to the singular individual. Liquid biopsy (LB), in this sense, presents new scenarios in personalized medicine through the study of molecules, factors, and tumor biomarkers in blood such as circulating tumor cells (CTCs), circulating tumor DNA (ctDNA), exosomes and circulating tumor microRNAs (ct-miRNAs). Moreover, its easy application and complete absence of contraindications for the patient make this method applicable in a great many fields. Melanoma, given its highly heterogeneous characteristics, is a cancer form that could significantly benefit from the information linked to liquid biopsy, especially in the treatment management. In this review, we will focus our attention on the latest applications of liquid biopsy in metastatic melanoma and possible developments in the clinical setting.

## 1. Introduction

The past few decades of research have achieved milestones in cancer diagnosis, management, and treatment. It is known that a tumor mass is not a single entity, but rather it is made up of sets of cells, very different from each other, communicating with the surrounding microenvironment that influences its growth, progression, and response to treatments [1]. New technologies and omics approaches have enabled a deep knowledge of cancer that translates, of course, into targeted therapies tailored to the individual patient. Precision medicine, indeed, aims to identify the specific characteristics of the individual to be translated into more effective therapies over time [2]. The in-depth study of the tumor is, therefore, crucial for patient management in terms of predicting responses to therapy and in cases in which treatment changes are needed. Indeed, the tumor is constantly evolving, and this is one of the reasons behind the failure of many therapeutic treatments and the establishment of secondary resistance. In this scenario, liquid biopsy (LB) proves to be a strong ally for the clinician who can monitor the progress of response to a treatment and any recurrence of disease, and decide to undertake treatments more suitable for the specific patient [2]. Extremely simple to perform and free of contraindications, LB has significant advantages compared with traditional biopsy which is extremely invasive and difficult to replicate over time [3].

Metastatic melanoma represents one of the tumors that could most benefit from the use of LB. Recommended current diagnostic approaches for patients with metastatic melanoma include ultrasound, magnetic resonance imaging (MRI), and positron emission tomography (PET)-CT which play important roles in tumor staging, surveillance, and assessment of therapeutic response [4]. The assessment of BRAF mutational status is also currently considered in metastatic melanoma diagnosis [5]. In the last decade, the assessment of other molecular pathways has been explored to evaluate genetic tumor evolution, for prognostic purposes, to monitor treatment response and acquired drug resistance [6]. Longitudinal analysis of LB samples is an emerging potential technology to dissect the complex clonal architecture of cancers without performing repeated and invasive tissue biopsies. Indeed, given its extremely heterogeneous characteristics, treatment of melanoma is still unresponsive in a significant number of patients due to the development of secondary resistance [7]. As demonstrated and discussed by several authors, LB can overcome the limits of traditional treatments. LB screening throughout immune [8] and/or target therapy [9] is crucial for patient monitoring in order to lead clinical decisions for personalized therapy.

The purpose of this review is to analyze the most interesting aspects of LB (in terms of methodologies and applications) and its implication in the detection, prognosis, and monitoring of metastatic melanoma.

## 2. Metastatic Melanoma

Although accounting for only 10% of the total, melanoma represents the deadliest form of skin cancer [10]. Melanoma development arises from atypically transformed melanocytes, commonly in the skin, in the setting of ultraviolet radiation (UVR) injury, and is most often caused by sun exposure and sunbeds. Sun exposure is the most significant environmental cause of skin cancer and continuous exposure to UVR is correlated with the occurrence of this disease. Other epidemiological risk factors such as pigmentation characteristics, high density of freckles, and a high number of naevi affect susceptibility to melanoma [11]. When not diagnosed early, the survival of patients with melanoma is drastically reduced, with only 15% of the total surviving 3 years after diagnosis [1,12]. Over the last 10 years, targeted therapies and immunotherapies have significantly improved responses and survival trends, changing the clinical management of patients with metastatic melanoma. Recently, several studies have focused on checkpoint inhibitor molecules that down-regulate immune responses. Immune checkpoint inhibitors against cyto-toxic T-lymphocyte-associated antigen 4 (CTLA-4) (Ipilimumab) and against programmed cell death protein 1 (PD-1) (Nivolumab) are currently used in advanced melanoma. Moreover, the discovery of BRAF mutations in several cancers including melanoma led to the development of BRAF kinase and MEK kinase inhibitors [13]. According to the current guidelines, BRAF-wildtype patients with advanced melanoma should either receive a dual therapy with ipilimumab plus nivolumab or a monotherapy with nivolumab or pembrolizumab (PDL-1 inhibitor), on the basis of clinical discretion. The same recommendation is valid for BRAF-mutant patients including an additional option with BRAF/MEK inhibitors [14,15]. Melanomas are histologically classified according to the tumor, node, and metastasis system (TNM) regulated by the American Joint Committee on Cancer (AJCC), which determines the stage of the tumor (T) through specific and universal characteristics such as tumor thickness, ulceration, and mitosis in lesions. Other parameters used for the classification of this pathology concern its ability to involve the lymph node system (N) and the possible presence of metastases (M) distant from the primary tumor [16,17,18].

The melanoma clinical stage depends on the involvement of the lymph node system and the infiltrating capacity of the disease; indeed, stage I–II–III melanomas include patients characterized by the absence or microscopic presence of lymph node and/or distal metastases whereas stage IV provides for the massive presence of distal metastases [17]. The less favorable prognoses are obviously linked to the more advanced stages of the disease. In recent years, many studies have shed light on the mechanisms underlying melanoma development and have shown that the melanocytes tumor transformation process is complex and multi-stage [19,20]. It is clear that the most benign lesions present the alteration of v-Raf murine sarcoma viral oncogene homolog B (BRAF) in the codon V600E (that is sufficient for the nevus formation), but for melanoma development, BRAF mutation is not sufficient because the disease progression is bound to concomitant alteration in other genes involved in the most important cellular processes [21,22]. Indeed, the benign nevi remain quiescent even for several years and, only after possible genetic mutations against target genes such as telomerase reverse transcriptase (TERT), cyclin-dependent kinase inhibitor 2A (CDKN2A), phosphatase and tensin homolog deleted on chromosome 10 (PTEN), neurofibromin 1 (NF1), and receptor tyrosin kinase (KIT), does neoplastic transformation begin. These genetic alterations are responsible for the uncontrolled activation of mithogen-activated protein kinase (MAPK) and phosphatidylInhositol3-kinase (PI3K) pathways that are physiologically involved in cell proliferation and survival [23]. It is certain that the MAPK pathway is the most dysregulated in melanoma and includes three major kinase families, i.e., MAPK kinase kinase, MAPK kinase, and MAPK, which activate and phosphorylate downstream proteins [24]. This pathway exhibits anomalies in many tumor contexts as it promotes proliferation and is involved in numerous treatment resistance processes [25,26]. Although less frequently, the PI3K pathway is also mutated in melanoma [27]. For example, PTEN (an important negative regulator of the PI3K pathway) in melanoma is frequently mutated/deleted and its loss of function is present and concomitant with BRAF mutations in approximately 44% of melanomas [28,29].

## 3. Liquid Biopsy (LB): An Overview

Precision medicine requires the study and analysis of a tumor in real time. Indeed, a tumor evolves over time, and its genetic, metabolomic, etc., characteristics can undergo profound changes [30]. This leads to variations in terms of disease progression and response to therapies. Traditional biopsies, performed invasively on tissues and organs, cannot meet (for patient management and cost reasons) the information requirements required to draw a map and determine tumor evolutions. Moreover, this method, which is still used for tumor diagnosis, is related to the performing time and provides a limited picture of the tumor, not taking into account tumor heterogenicity [31]. LB, on the other hand, presents new scenarios in real-life oncology, providing tumor assessment throughout a painless and feasible blood draw over time. LB is not only limited to blood but can also be represented by several other human fluids such as semen, urine, cerebrospinal fluid (CSF), saliva, pleural fluid, and ascites [32,33,34] with a biomarkers quantity largely linked to the localization of the primary tumor and metastasis [35]. For instance, CSF is used to monitor glioblastoma patients as it has a higher amount of ctDNA and CTCs than blood [34]. Urine, instead, is used in a new urine-based assay to detect single mutant molecules of fibroblast growth factor receptor 3 (FGFR3) that are indicative of bladder cancer, which represents a noninvasive tool of early-stage diagnosis [32]. Furthermore, to explore the potential of ctDNA as a biomarker for head and neck squamous cell carcinomas (HNSCC), Wang at al. studied ctDNA from the saliva of 93 HNSCC patients, showing that the sensitivity was site-dependent and most efficient for tumors in the oral cavity. Instead, in HNSCCs distal to the oral cavity (oropharynx, larynx, and hypopharynx) ctDNA was often detectable, but the frequency and the fraction of mutant alleles were considerably lower than those found in the oral cavity [33].

This review will focus on the analysis of circulating tumor cells (CTCs), circulating tumor DNA (ctDNA), exosomes, and circulating tumor microRNAs (ct-miRNAs) released by cancer cells in blood [36] (Figure 1).

An important LB benefit is the possibility of monitoring the tumor mass. Indeed, monitoring makes it possible to assess changes (of various types) related to detectable tumor components in fluids over time. All tumor products can fluctuate in their concentrations and mutational and molecular status depending on the characteristics of the tumor cells and the stage of the disease (Table 1).

### 3.1. Circulating Tumor Cells (CTCs)

CTCs are tumor cells originating from the primary site of the malignancy or from metastasis, which give an instantaneous picture of the disease [58,59]. These cells are present in the bloodstream in very low concentrations (1–10 cells/milliliter of blood) but, compared with standard tissue biopsies, are able to provide fundamental information on inter- or intra-tumoral heterogenicity from genetic aberrations and transcriptional and epigenetic dysregulation in that a small amount of tissue may not be representative of the whole tumor [60,61,62,63,64]. CTCs are critical for tumor analysis as several studies have shown that blood concentration changes more significantly and in closer correlation with the tumor origin than other tumor markers [62,65,66]. The detection and isolation methods of CTCs are different but are all essentially based on the physical and biological characteristics of the cells [67]. The EPithelialImmunoSPOT (EPISPOT) assay (which exploits tumor expression of specific molecules) allows blood detection of even a single cell and has produced significant results in terms of cell concentration assessment in several solid tumors [68,69]. The assay discriminates tumor cells from “normal” cells by selectively recognizing the expression of the epithelial cell adhesion molecule (EpCAM) [70]. Unlike other solid tumors, such as adenocarcinomas of the colon and pancreas and hormone-refractory adenocarcinomas of the prostate, melanoma has non-high levels of EPCAM protein [71,72]. For example, Odashiro and collaborators, in a study conducted on 25 patients with uveal melanoma, did not any detect positivity for EPCAM expression in the samples analyzed [73]. Unlike other solid tumors, melanoma has non-high levels of EPCAM protein. Therefore, it was necessary to identify other molecules to discriminate circulating melanoma cells, and specifically, a new EPISPOT assay based on S-100 protein recognition (S100-EPISPOT) was developed [71,74].

#### CTCs and Melanoma

In recent years, the development of precision and personalized medicine has required the need to capture pictures of tumor progression. Highly heterogeneous tumors, such as metastatic melanoma, undergo profound changes dictated by intrinsic cell alterations or drug treatments. Therefore, the analysis of tumor behavior through the study of CTCs appears increasingly cutting edge. Many studies have focused their attention on melanoma CTCs.

In 2013, Karakousis and collaborators highlighted the need to monitor patients with metastatic melanoma, even during therapy, to correct the target as soon as favorable conditions changed. The authors described that in 101 patients with stage IV melanoma, 26% of them had values ≥2 of CTCs at baseline. The account of CTCs was evaluated with the CellSearch CTC enumeration system, which used Melcam and high-molecular-weight melanoma-associated antibody (HMW-MAA), 2 markers expressed in up to 80% of metastatic melanoma lesions. The number of CTCs was significantly correlated with overall survival (OS) and thus could be considered an important prognostic factor. Indeed, during treatment, patients with no significant decreases in the number of CTCs (≥2) had a worse median OS (7 vs. 10 months) than, in contrast, the group of patients with decreased values of CTCs (<2) [37]. A more recent study investigated the prognostic role of CTCs in 93 patients with stage IV metastatic melanoma, whose CTCs baseline levels were assessed by an immunomagnetic system. All patients that presented values ≥ 1 were considered positive and eligible. Confirming the findings of the previously described study, a significant correlation was found between baseline levels of CTCs and the progression of disease (PD) within 180 days of detection. Therefore, in metastatic melanoma, CTCs can be considered as possible biomarkers that can identify patients with a higher risk of PD [38]. In stage III melanoma patients, the association between the CTCs and disease relapses was evaluated in a recent clinical study. In 243 patients, CTCs detection was significantly associated with shorter relapse-free survival (RFS) [39].

In addition to their prognostic role in survival and PD, CTCs can also provide accurate and real-time information in response to treatments. Kiniwa et al. evaluated the role of CTCs as biomarkers of treatment response in BRAF-mutated melanoma. Although performed in a limited number of patients, the study showed a fluctuating trend in CTCs levels in 4 of 5 patients with grade IV metastatic melanoma treated with BRAF and MEK inhibitors. This finding, which also takes into account the high heterogenicity of melanoma cells, points out the correlation between the number of CTCs and the response to treatment [40]. Like target therapies, immunotherapy is showing important results in melanoma treatment and CTCs appear to be a predictive biomarker of response to this type of treatment.

Khattak and collaborators recently showed, in patients with advanced melanoma, a significant correlation between the levels of PDL1-rich CTCs and the response to pembrolizumab. Indeed, patients with CTCs/PDL1+ had favorable effects in terms of progression-free survival (PFS) compared with the same patients with low circulating levels of PDL1 [41]. Constant monitoring of CTCs could be a valuable support for clinicians in screening patients going through therapies, thus avoiding unnecessary and harmful treatments. For this purpose, for example, the development of a 19-gene signature for melanoma CTCs has enabled early and rapid assessment of the response to immunotherapies. Indeed, changes in this gene signature after treatment provide important data about the response to therapies, even in the long term [42].

### 3.2. Circulating Tumor DNA (ctDNA)

The tumor presence in the bloodstream is represented not only by cells but also by their waste products, such as tumor DNA. Indeed, tumor can release genetic material for a variety of reasons, mainly in the case of programmed death by apoptosis or cell necrosis [75,76]. At the same time, the high genomic instability that characterizes cells undergoing rapid division, as in the metastatic process, can further result in the release of DNA into the circulation [77,78]. Circulating tumor DNA (ctDNA) is essentially represented by small fragments of genetic material (<166 bp) that undergoes changes in terms of blood concentrations during PD or after drug treatments. ctDNA represents only the smallest fraction of all circulating free DNA (cfDNA) released even by non-tumor cells. The percentage of ctDNA varies from one patient to the other and using a quantitative PCR-based approach has determined that the proportion of tumor DNA varies between 10% and 90% of total cfDNA, with the highest proportions in the samples with low overall cfDNA levels [79]. In contrast, an earlier study reported much lower proportions of tumor DNA (between 0.2% and 10%) [80]. The concentration of cfDNA in the serum of cancer patients is about 4 times higher than that of healthy controls and is observed in healthy subjects at concentrations between 0 and 100 ng/mL of blood with an average of 30 ng/mL. However, in cancer patients a fraction of this total cfDNA contains tumor-specific somatic alterations derived from ctDNA and the concentration in plasma or in serum varies between 0 and 1000 ng/mL, with an average of 180 ng/mL [81]. Isolation methods are based on the detection of mutations (mt), rearranged genomic sequences, copy number variations (CNV), microsatellite instability (MSI), amplified sequences, loss of heterozygosity (LOH), DNA methylation (DNAm), and the degree of integrity [82,83,84]. There are several kits available for isolating ctDNA based on different techniques of processing plasma including affinity column, magnetic bead, polymer, and phenol-chloroform methods. These methods vary in their ability to purify fragments of different sizes. Consequently, they may change the total quantity of cfDNA isolated and skew the fraction of ctDNA [85]. Highly sensitive and specific methods are currently used to detect ctDNA, such as droplet digital PCR (ddPCR) or next generation sequencing (NGS) [86].

In metastatic melanoma patients, ctDNA detection promotes the identification of typical mutations (i.e., BRAF and Neuroblastoma RAS Viral Oncogene Homolog (NRAS)) and epigenetic markers such as methylated DNA [47,87]. Indeed, DNA methylation of specific genes, such as Tissue Factor Pathway Inhibitor 2 (TFPI2), has been shown to be correlated with advanced melanoma [88]. Several studies have shown ctDNA fluctuations in different tumor contexts, such as in pancreatic cancer and BRAF-mutated metastatic melanoma [89,90]. In recent years, ctDNA as a biomarker of treatment response and PD has assumed a key role in cancer patient monitoring, providing a real-time view of the tumor and allowing for the assessment of changes in the disease mutational structure [89,90,91,92]. Furthermore, it has long been known that ctDNA can influence the course of tumor progression because it is a biologically active product. An interesting study conducted pre-clinically in patients with colon cancer showed that the presence of ctDNA in the supernatant could induce murine NIH3T3 cells toward malignant transformation [93].

#### ctDNA and Melanoma

The high prognostic potential of ctDNA has increased studies in recent years on its role as an indicator of disease and response to treatment in various cancer settings, particularly in melanoma. To date, although many steps have been taken, there is still no clear and specific evidence on the prognostic validity of ctDNA due to studies with sometimes conflicting results.

One of the most frequent sites of melanoma metastasis is the brain and the OS of patients characterized by this metastatic site is about 4–6 months [94]. In an interesting study, Lee and collaborators evaluated the potential role of ctDNA for surveillance and monitoring of the response to systemic therapy in patients with melanoma brain metastases. Circulating levels of BRAF, NRAS, and c-KIT mutations were assessed in 72 patients with brain metastases from melanoma and who were undergoing anti-PD1 treatment. The study showed that patients with lower ctDNA values (at baseline and/or during treatment) had better OS [43]. Studies involving the analysis of ctDNA are developing rapidly. Recently, another research group focused on creating a data platform for LB and correlated ctDNA levels and patient survival. Specifically, the authors selected 19 subjects with stage III and IV melanoma and, after validating their mutational status matched in ctDNA, observed that patients with higher levels of circulating genetic material had significantly decreased PFS [44]. ctDNA levels can be considered a good marker of response to treatments, especially those with a nonimmediate outcome, such as immunotherapies, and can therefore help in stratifying patients for treatment. It was observed that melanoma patients characterized by lower pre-treatment (anti-PD1) ctDNA levels had a longer PFS. The group of patients with higher blood concentrations of ctDNA had better survival ranges only when treated with the combination of anti-PD1 and CTLA-4 inhibitors [45].

With completely opposite results, a very recent study, in patients with metastatic melanoma, had compared the function of ctDNA as a PD indicator with radiomics imaging. The authors analyzed ctDNA, obtained at the stage of PD, from 108 patients with melanoma, and 66 patients were monitored after response to therapy. Although well-articulated, the results obtained from the retrospective and prospective study did not show good efficacy of ctDNA in detecting PD in melanoma, when compared with analysis of standard positron emission tomography imaging. ctDNA was detected in only 62% of patients at the time of PD and only a non-significant number of patients showed changes in ctDNA blood levels between the onset of PD and treatment response [46].

### 3.3. Exosomes

Exosomes are small (30–200 nm) extracellular vesicles (EV) secreted by all cells for intercellular communication by transferring functional proteins, metabolites, and nucleic acids [95,96]. Initially considered a mechanism by which cells expel waste material, it is now known that exosomes play an important role in cellular communication and influence several cellular processes, such as immune response and maintenance of cell stem state, and are obviously involved in tumor mechanisms [97,98,99,100]. Indeed, they are central players in the survival and growth of the primary tumor by promoting exchanges and interconnections between tumor and non-tumor cells; they stimulate extracellular matrix remodeling by inducing tumor migration and invasion; and they promote angiogenesis, which is necessary for the metastatic process [101].

The exosome isolation methods differ depending on the material they contain. In a recent study, Mondal and colleagues discriminated tumor exosomes from other EVs using a size-exclusion chromatography (SEC) followed by an immunoaffinity approach, based on an anti-chondroitin sulfate peptidoglycan 4 (CSPG4) monoclonal antibody specific for an epitope expressed only by melanoma cells [102]. Although it is not very easy to discriminate vesicles produced by “normal” cells from those released by tumor cells, the great abundance of these structures in circulation gives them the function of possible biomarkers in different tumor contexts [103,104,105,106]. Melanoma cells not only use molecules released via exosomes to influence the surrounding microenvironment in their favor, they can also induce the reprogramming of fibroblasts into cancer-associated fibroblasts (CAFs) via Gm26809 delivery and thereby stimulate tumor progression and migration [107]. In addition, melanoma stimulates the stromal release of pro-inflammatory cytokines, such as interleukins 6 and 8 (IL-6 and IL-8), resulting in the establishment of a pro-tumorigenic microenvironment. Therefore, pharmacological action targeted toward these elements could open up new therapeutic scenarios [108].

#### Exosomes and Melanoma

Several studies have investigated the possible tumor biomarker role of exosomes. There is evidence that patients with melanoma have higher levels of protein content of exosomes than healthy subjects, even in non-active disease settings.

A 2020 study conducted in 19 melanoma patients and 6 healthy donors found that the expression of immunosuppressive proteins was higher in exosomes isolated from the peripheral blood of melanoma patients than in those of healthy donors. This could be related to the ability of the tumor to suppress the immune response and inhibit the action of immunotherapy agents [48]. The ability to monitor disease evolution over time is one of the foundations on which precision medicine is based; it focuses on intervening treatments and improving drug performance.

Cordonnier and collaborators analyzed exosomal levels of PDL-1 (with immunosuppressive properties) in 100 melanoma patients and observed a significant increase in this circulating protein compared with healthy controls (64.26 pg/mL vs. 0.1 pg/mL). The authors also pointed out an increased sensitivity of LB compared with standard tissue biopsies because serum levels of PDL-1 were detected in all patients analyzed, whereas only 67% of tissue biopsies were PDL-1 positive. Changes in the serum concentration of PDL-1 could, therefore, be used as a biomarker of treatment response and clinical outcome [49]. Exosomal PDL-1 immunosuppressive action has been extensively investigated in melanoma in several other studies. Indeed, PDL-1, as previously described, can not only lead to immune evasion, but can also antagonize anti-PD-L1 therapy by binding to the antibody itself [103,109].

Alegre and collaborators underlined the possible prognostic role of exosomes with a study that examined the presence of biomarkers such as melanoma inhibitory activity (MIA), S100B, and tyrosinase-related protein 2 (TYRP2) in exosomes obtained from the serum of healthy donors, metastatic melanoma, and melanoma disease-free patients. The study showed that melanoma patients had significantly higher S100B and MIA exosomal concentrations than the negative controls analyzed. Furthermore, patients with exosomal concentrations of MIA greater than 2.5 μg/L had a shorter PFS than those with a lower level (4 vs. 11 months; *p* < 0.05); however, TYRP2 showed no significant differences within the three study groups [50]. The study of exosomes may also provide important information for the development of new therapeutic strategies, especially in melanoma where the microenvironment strongly conditions the disease course. Indeed, a very recent study has shown that melanoma exosomes induce the pro-inflammatory function of cancer-associated fibroblasts (CAFs) by stimulating the production of cytokines such as IL-6 and IL8. These factors, therefore, could be considered a valuable therapeutic target [108]. Exosomes produced by melanoma cells markedly influence progression and the ability of the tumor mass to evade the immune system. The presence of exosomes in vitro results in the establishment of a microenvironment consisting mainly of type 2 macrophages and suppressive dendritic cells, suggesting the triggering of immunosuppressive mechanisms [110]. Melanoma cells secrete into the bloodstream EVs as well as exosomes containing microRNAs [111]. Most recently, Sabato et al. characterized plasma EVs associated with miRNAs (pEV-miRNAs) profiles from metastatic melanoma patients providing a melanoma-specific ct-miRNAs signature. Through a miRNA bioinformatic analysis they identified a panel of four pEV-miRNAs, namely, miR-412-3p, miR-507, miR-1203, and miR-362-3p, with a high diagnostic power [51].

### 3.4. Circulating Tumor microRNA (ct-miRNAs)

MicroRNAs are small endogenous (19–22 nt) single-stranded non-coding RNA molecules able to modulate gene expression at the post-transcriptional level, with a key role in multiple cellular processes and regulatory pathways [112]. MiRNAs can be secreted into biological fluids, such as blood, semen, urine, and cerebrospinal liquid, after cell death processes or to selectively mediate intercellular signaling [113].

Circulating tumor miRNAs (ct-mirRNAs) are more stable compared with other extracellular RNAs: they are complexed with proteins or lipoproteins such as Argonaute 2 (AGO2), or packaged inside EVs which protect them against RNase activity [114]. Despite high stability in blood, their low concentration and lack of consistent processes for collection make the pre-analytical phase of ct-miRNAs detection a critical step to minimize analytical variabilities [115]. Commercially available kits for RNA extraction and real-time quantitative reverse transcription polymerase chain reaction (qRT-PCR) for quantification are the gold standard in ct-miRNAs assessment [116].

Due to their presence in readily accessible body fluids, since 2008, the potential role of ct-miRNAs as promising diagnostic, prognostic, and predictive biomarkers in solid cancer has been investigated [116,117,118,119,120,121]. Many ct-miRNAs are found to be up- or down-regulated in the blood of cancer patients [122]. Up-regulation of miR-221, involved in cell cycle regulation and proliferation, has been observed in a large number of tumors such as glioblastoma [123], lung cancer [124], breast cancer [125], thyroid papillary carcinoma [126], hepatocellular carcinoma [127], and melanoma [114]. Down-regulation of miR-192, miR-194, and miR-215 prevent apoptosis in multiple myeloma [128].

#### ct-miRNAs and Melanoma

In the last few years ct-miRNAs have been proposed as promising non-invasive biomarkers for melanoma. Several studies investigated ct-microRNAs functional role in melanoma, as extensively summarized by Ghafouri-Fard et al., [129]. For instance, there is evidence of their involvement in the regulation of MAPK/ERK, PI3K/PTEN/AKT and NF-kb signaling pathways, involved in several cellular processes foundamental for tumor biogenesis, such as proliferation, migration and invasion [24,130,131]. However, the underlying mechanisms are not clarified completely [129]. This section will focus on the employment of ct-miRNAs as promising quantitative non-invasive biomarkers for melanoma. Particularly, significative alterations in ct-miRNAs expression are crucial to identified melanoma patients and stratified them as metastatic and non-metastatic [132]. Most recently Ruggiero et al. showed that specific ct-miRNAs (miR-579-3p and miR-4488) can be used to predict response to target therapy. Retrospective analysis was carried out in 70 serum samples derived from BRAF-mutated melanoma patients treated with MAPKi in order to discover a possible “mini-signature” identifying ones who could benefit from therapy [52].

Instead of using in isolation, panels of miRNAs may result in increased sensitivity and specificity. Van Laar et al. used a combination of 38 miRNAs (MEL38) to differentiate healthy controls from patients who had stage I-IV melanoma [53]. Stark and collaborates identified a panel of seven melanoma-related biomarkers (MELmiR-7) able to detect an increase in tumor burden in 100% of cases analyzed. The ‘MELmiR-7’ panel characterizes overall survival of melanoma patients better than both serum LDH and S100B [54]. An other study highlighted higher levels of miR-199a-5p in advanced stages of desease promoting melanoma metastasis and angiogenesis. Moreover miR-199a-5p in combination with up-regulated miR-877-3p, miR-1228-3p, miR-3613-5p and down-regulated miR-182-5p, correlated with higher melanoma stages at the time of primary melanoma excision [55].

Due to its immunogenic profile and higher mutational burden, melanoma represent an ideal model to investigate the interplay between cancer cells and immune cells. Jorge and colleagues identified a correlation between ct-miRNAs in metastatic melanoma samples and poor clinical outcome, associated to immune evasion and tumor microenvironment response [56]. Bypassing host systemic immune control, metastatic melanoma can develop resistance to immunocheckpoint inhibitors (ICIs). Recently a prospective pilot study demonstrated the potential role of ct-miRNAs as blood biomarkers for stage III and IV melanoma patients in assessing ICIs responses. MiR-1234-3p, miR-4649-3p and miR-615-3p were significantly increased in post-treatment samples from stage IV non-responder patients, while miR-4649-3p, miR-1234-3p and miR-615-3p decreased in post-treatment samples of stage IV patients who had a complete response during ICIs treatment. When comparing stage III responders versus non-responders, only miR-3197 was found differentially expressed [57].

## 4. Conclusions

Metastatic melanoma could significantly benefit from LB applications. Given its high heterogeneity, melanoma treatment is in many cases unresponsive due to intrinsic resistances or to the development of secondary resistances [7]. In this sense, LB could be an advantage method not only to diagnosis and prognosis but above all for patient management in order to monitor treatment response, relapse, and progression of disease, presenting new scenarios toward personalized medicine [2]. LB provides clinical information obtained from the analysis of CTCs, ctDNA, exosomes, and ct-miRNAs. Moreover, LB enables a less invasive approach, is more cost-effective, easier to perform, and is performed with high throughput testing compared with traditional biopsy. To date, there are currently no clinically validated stage-specific liquid biomarkers to stratify melanoma patients, and thus further investigations are needed.

## Figures and Tables

**Figure 1 ijms-24-04014-f001:**
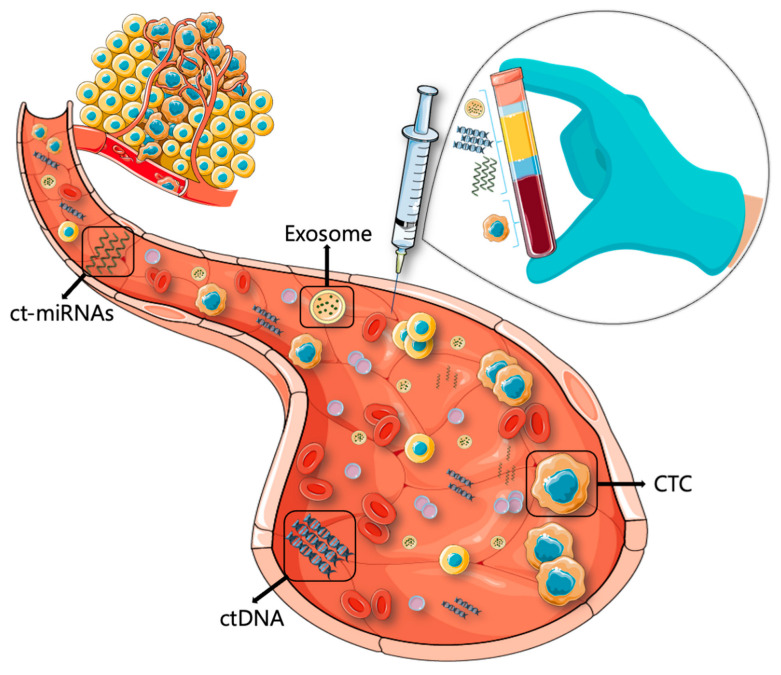
A schematic view of liquid biopsy. Blood collected from cancer patients contains circulating tumor cells (CTCs), circulating tumor DNA (ctDNA), exosomes, and circulating tumor microRNA (ct-miRNA). These biomarkers could provide real-time information on tumor progression, prognosis, and treatment response.

**Table 1 ijms-24-04014-t001:** Most important tumor alterations investigable in circulating biomarkers by sequential LB analysis.

LB Biomarkers	Type of Evaluation	Results	Reference(s)
**CTCs** 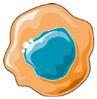	**Enumeration**	Correlation between increased CTCs number and worse OS.	[37]
Significant correlation among baseline levels of CTCs and PD.	[38]
Association of CTCs detection with a shorter RFS.	[39]
Correlation between CTCs number and BRAF and MEK inhibitors responses.	[40]
**Molecular ** **characterization**	Favorable effects in CTCs/PDL1+ patients in terms of PFS.	[41]
Early assessment of response to immunotherapies by means of a 19-gene signature.	[42]
**ctDNA** 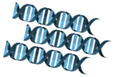	**Mutation ** **detection**	Association between lower levels of circulating BRAF, NRAS, and c-KIT mutations at baseline and/or during anti-PDL1 treatment and better OS.	[43]
**Quantification**	Higher levels of circulating genetic material correlated with significantly decreased PFS.	[44]
Correlation among higher blood concentrations of ctDNA and better survival ranges withanti-PD1/CTLA-4 inhibitors treatment.	[45]
Detection of changes in ctDNA blood levels between the onset of PD and treatment response.	[46]
**DNA ** **methylation**	Correlation between methylation of TFPI2 with metastatic melanoma.	[47]
**Exsosomes** 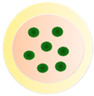	**Proteomic ** **characterization**	Higher expression of immunosuppressive proteins in exosomes from the blood of melanoma patients than those of healthy donors.	[48]
Significant increase in circulating PDL1 compared with healthy controls.	[49]
Significant association between high S100B and MIA exosomal concentrations and shorter PFS.	[50]
**Circulating exsosomal miRNA**	High-power diagnostic assessment of apEV-miRNAs panel (miR-412-3p, miR-507, miR-1203, and miR-362-3p).	[51]
**ct-miRNA** 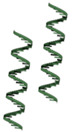	**Quantification**	Prediction of target therapy responses with specific ct-miRNAs (miR-579-3p and miR-4488).	[52]
Stratification of stage I–IV melanoma patients by a combination of 38 miRNAs (MEL38).	[53]
Detection of tumor burden increase through a panel of seven melanoma-related biomarkers (MELmiR-7).	[54]
Correlation between levels of miR-199a-5p, miR-877-3p, miR-1228-3p, miR-3613-5p, miR-182-5p, and higher melanoma stages.	[55]
Correlation between ct-miRNAs in metastatic melanoma samples and poor clinical outcome, associated to immune evasion and tumor microenvironment response.	[56]
Prediction of treatment response in melanoma stage IV patients with a panel of ct-miRNA (miR-1234-3p, miR-4649-3p, and miR-615-3p).	[57]

## Data Availability

Not applicable.

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
