# Peer review of "Metastatic Melanoma: Liquid Biopsy as a New Precision Medicine Approach"

_ijms, 2023, doi:10.3390/ijms24044014_

Round 1
Reviewer 1 Report
The review of Elena Ricciardi and collaborators is well written and interesting. The field of melanoma is a current problem in health care. Works on miRNAs are properly discussed.
Strong point: Clear cut literature survey.
Weak point: A previous review on a very similar topic has been published.
Fattore L, Ruggiero CF, Liguoro D, Castaldo V, Catizone A, Ciliberto G, Mancini R. The Promise of Liquid Biopsy to Predict Response to Immunotherapy in Metastatic Melanoma. Front Oncol. 2021 Mar 18;11:645069. doi: 10.3389/fonc.2021.645069. PMID: 33816298; PMCID: PMC8013996. I suggest to authors to quote and discuss this paper.
Minor alteration
In the “Conclusions section”, the sentence: “Indeed, it is possible
observe real-time changes in the tumor on: CTCs enumeration and molecular characteri-
zation; ctDNA quantification, mutations detection and DNA methylation investigation;
exosomes proteomic characterization; ct-miRNAs quantification” may be suppressed or rewritten as it is only giving the information that real-time changes can be observed with Liquid biopsy.
Author Response
We thank the reviewer for positive comments on the manuscript. We have proceeded to cite in the text the paper recommended (Fattore L. et al.) by the reviewer and removed the sentence mentioned in the conclusion section.

Reviewer 2 Report
Review Report for the Manuscript “Metastatic Melanoma: Liquid Biopsy as a new precision medicine approach.”
Rating the Manuscript
Quality of Presentation: Is the article written in an appropriate way? Are the highest standards for presentation of the results used?
Yes, the article is written well. Representation could be improved. Specially authors could use tables in order to summarize the information in the manuscript.
Interest to the Readers: Are the conclusions interesting for the readership of the Journal? Will the paper attract a wide readership, or be of interest only to a limited number of people? (Please see the Aims and Scope of the journal)
Yes, this would be a great article and contain important information about liquid biopsy markers. This would be an interesting read for the researchers in the cancer research field.
English Level: Is the English language appropriate and understandable?
Yes, English language in the manuscript is appropriate and understandable.
Overall Recommendation: Accept after Minor Revisions
Given below are the comments for each section of the manuscript.
Author list
If the authors Elena Ricciardi, Elena Giordani, Antonino Guerrisi and Fabio Valenti equally contributed to the manuscript, authors could group them together in the beginning of the authors list. Here, two of the authors are listed in the beginning of the manuscript and the other two are listed at the end of the author list.
Abstract
The abstract is written and summarizes the content of the manuscript.
Line 31: “Liquid biopsy (LB) has, in this sense, opened new scenarios toward personalized medicine through the study of molecules, factors and tumor biomarkers in the blood.”
Here authors could mention what tumor biomarkers are specifically discussed in the manuscript.
1.Introduction
Authors could discuss the currently used methods diagnosis of Metastatic Melanoma, drawbacks of these methods and the need for using liquid biopsy markers for the diagnosis purposes.
2.Metastatic Melanoma
In the beginning of this section, authors could briefly discuss about the causes of metastatic melanoma and any treatment that’s available.
Line 83: “In recent years, many studies have shed light on the mechanisms underlying melanoma development and have shown that the melanocytes tumor transformation process is complex and multi-stage.”
Authors need to add few references here.
Line 95: “These genetic alterations are responsible of uncontrolled activation of mithogen activated protein kinase (MAPK) and phosphatidylInhositol3-kinase (PI3K) pathways that are physiologically involved in cell proliferation and survival”.
Authors could briefly discuss about these pathways.
3. Liquid Biopsy (LB): an overview
Figure 1: Nice figure, this summarizes the content of the manuscript and make it easy to follow the manuscript.
In this section authors could discuss different types of bodily fluids from which these liquid biopsy markers can be isolated. They could explain the abundance of these markers in these bodily fluids and how they can be used for diagnosis of different types of cancer.
Figure 2: It would be great if the authors could include a table with all the information shown in the figure along with few references for each application explained in this figure.
3.1. Circulating Tumor Cells (CTCs)
Line 127: “These cells are present, in the bloodstream, in very low concentrations (1-10 cells/ milliliter of blood) but, compared with standard tissue biopsies, are able to provide fundamental information on tumor heterogenicity.”
Discuss the advantages of using CTCs compared to tissue biopsy.
Line 138: “Unlike other solid tumors, melanoma has non-high levels of EPCAM protein. Therefore, it was necessary to identify other molecules to discriminate circulating melanoma cells, and specifically, a new EPI-SPOT assay based on S-100 protein recognition (S100-EPISPOT) was developed”.
Compare the expression levels of EpCAM in melanoma cells to the other cancer cells. State a value so it’s easier to understand why EpCAM cannot be used as a marker for melanoma.
3.1.1. CTCs and melanoma
Line 152: “The authors described that in 101 patients with stage IV melanoma, 26% of them had values ≥ 2 of CTCs at baseline.”
Authors need to discuss what methods are being used to isolate the CTCs in these studies.
3.2. Circulating Tumor DNA (ctDNA)
Line 197: “ctDNA represents only the smallest fraction of all circulating free DNA (cfDNA) released even by non-tumor cells.”
What is the percentage of ctDNA from total cfDNA? And what's the abundance of ctDNA in blood one milliliter of blood?
Line 198: “Isolation methods are based on the detection of: mutations (mt), rearranged genomic sequences, copy number variations (CNV), microsatellite instability (MSI), amplified sequences, loss of heterozygosity (LOH), DNA methylation (DNAm) and the degree of integrity.”
Authors could briefly explain about the methods used isolating/purifying ctDNA from blood. Are there any commercial kits available for isolating ctDNA?
Line 208: “In recent years, ctDNA as a biomarker of treatment response and PD has assumed a key role in cancer patient monitoring, providing a real time view of the tumor and allowing for the assessment of changes in the disease mutational structure.”
Although the authors mention that studies in recent years, the reference 45 is from 1977.
3.3. Exosomes
Line 254: “Cell communication can occur through different ways, such as cell-to-cell contact or exposure to paracrine factors. Especially the paracrine release of molecules influences the entire cell life cycle and the potential response to drug treatments.”
Authors need to discuss more information about exosomes. They could discuss about facts like biogenesis and size distribution of exosomes. Also, they need to mention that exosomes are a sub population of extracellular vesicles.
Line 265: “In the melanoma case, to discriminate tumor exosomes, a recent study used an immunoaffinity approach based on an anti- chondroitin sulfate peptidoglycan 4 (CSPG4) monoclonal antibody, specific for an epitope expressed only by melanoma cells”.
How did they specify those were exosomes, but not any other type of extracellular vesicles?
3.4.1. ct-miRNAs and melanoma
Line 354: “Sabato et al. characterized plasma EVs associated miRNAs (pEV-miRNAs) profiles from metastatic melanoma patients providing a melanoma-specific ct-miRNAs signature. Through a miRNA bioinformatic analysis they identifed a four pEV-miRNAs panel, namely miR-412-3p, miR-507, miR-1203 and miR-362-3p with a high diagnostic power”.
If these are EV associated miRNA can they be considered as circulating tumor micro RNA? Did they isolate EVs and then extracted the miRNA? Then, I think this information is more related to the exosome section.
References
There are references that are more than 10 years old. In the abstract authors state that they focus on latest publications. Therefore, authors need to replace these old references with new references.
References: 9, 10, 16, 20, 21, 26, 27, 36, 37, 43, 45, 76, 79, 81, 83, 84, 85, 86 and 87

Author Response
We thank the reviewer for considering the manuscript interesting and valid for publication. We have made all requested changes in the text. In addition, the bibliography was appropriately edited as requested by the reviewer. In the appropriate section, the authors have been listed in a form that indicates the co-first and co-last of the manuscript.
